# Differential Thermotolerance Adaptation between Species of *Coccidioides*

**DOI:** 10.3390/jof6040366

**Published:** 2020-12-14

**Authors:** Heather L. Mead, Paris S. Hamm, Isaac N. Shaffer, Marcus de Melo Teixeira, Christopher S. Wendel, Nathan P. Wiederhold, George R. Thompson, Raquel Muñiz-Salazar, Laura Rosio Castañón-Olivares, Paul Keim, Carmel Plude, Joel Terriquez, John N. Galgiani, Marc J. Orbach, Bridget M. Barker

**Affiliations:** 1Pathogen and Microbiome Institute, Northern Arizona University, Flagstaff, AZ 86011, USA; hlb82@nau.edu (H.L.M.); Paul.Keim@nau.edu (P.K.); 2Department of Biology, University of New Mexico, Albuquerque, NM 87131, USA; psh102@unm.edu; 3School of Informatics, Computers, and Cyber Systems, Northern Arizona University, Flagstaff, AZ 86011, USA; ins2@nau.edu; 4Faculty of Medicine, University of Brasilia, Brasilia 70000-000, Brazil; marcus.teixeira@nau.edu; 5Department of Medicine, University of Arizona, Tucson, AZ 85721, USA; cwendel@aging.arizona.edu; 6Department of Pathology and Laboratory Medicine, University of Texas Health Science Center at San Antonio, San Antonio, TX 77030, USA; wiederholdn@uthscsa.edu; 7Departments of Internal Medicine Division of Infectious Diseases, and Medical Microbiology and Immunology, University of California-Davis, Sacramento, CA 95616, USA; grthompson@ucdavis.edu; 8Laboratorio de Epidemiología y Ecología Molecular, Escuela Ciencias de la Salud, Universidad Autónoma de Baja California, Unidad Valle Dorado, Ensenada 22890, Mexico; ramusal@uabc.edu.mx; 9Department of Microbiology and Parasitology, Universidad Nacional Autónoma de Mexico, Ciudad de México 04510, Mexico; lrcastao@unam.mx; 10Northern Arizona Healthcare, Flagstaff, AZ 86001, USA; CPlude@tgen.org (C.P.); Joel.Terriquez2@nahealth.com (J.T.); 11Valley Fever Center for Excellence, University of Arizona, Tucson, AZ 85721, USA; spherule@arizona.edu (J.N.G.); orbachmj@email.arizona.edu (M.J.O.); 12School of Plant Sciences, University of Arizona, Tucson, AZ 85721, USA

**Keywords:** coccidioidomycosis, fungal pathogen, phenotypic variation, growth rate, valley fever

## Abstract

Coccidioidomycosis, or Valley fever, is caused by two species of dimorphic fungi. Based on molecular phylogenetic evidence, the genus *Coccidioides* contains two reciprocally monophyletic species: *C. immitis* and *C. posadasii*. However, phenotypic variation between species has not been deeply investigated. We therefore explored differences in growth rate under various conditions. A collection of 39 C. posadasii and 46 C. immitis isolates, representing the full geographical range of the two species, was screened for mycelial growth rate at 37 °C and 28 °C on solid media. The radial growth rate was measured for 16 days on yeast extract agar. A linear mixed effect model was used to compare the growth rate of *C. posadasii* and *C. immitis* at 37 °C and 28 °C, respectively. *C. posadasii* grew significantly faster at 37 °C, when compared to *C. immitis*; whereas both species had similar growth rates at 28 °C. These results indicate thermotolerance differs between these two species. As the ecological niche has not been well-described for *Coccidioides* spp., and disease variability between species has not been shown, the evolutionary pressure underlying the adaptation is unclear. However, this research reveals the first significant phenotypic difference between the two species that directly applies to ecological research.

## 1. Introduction

Coccidioidomycosis, or Valley fever, is an environmentally acquired disease caused by inhalation of arthroconidia of dimorphic fungi belonging to the genus *Coccidioides*. In the environment, the fungi grow as filamentous mycelia, alternate cells of which autolyze and become fragile, leaving intact asexual arthroconidia that may disperse via wind or soil disruption. If inhaled by a susceptible host, an arthroconidium switches to a host-associated lifecycle and develops into a specialized infectious structure called a spherule. Subsequently, the host’s immune system either represses spherule replication or the host succumbs to the illness [1,2]. It is thought that symptomatic infection occurs in approximately 40% of human patients, who exhibit a broad spectrum of clinical symptoms, ranging from acute self-limited pneumonia, fibrocavitary chronic pulmonary infection, or hematogenous spread to extrapulmonary locations (i.e., disseminated infection) [3]. By one estimate, there are 146,000 new symptomatic U.S. coccidioidal infections each year [4] although the reported cases are substantially lower [5].

Coccidioidomycosis is caused by two species, *C. immitis* and *C. posadasii*. Genetic analysis of multiple molecular markers has defined two monophyletic clades [6]. Subsequent population genetic/genomic studies revealed that *C. immitis* is composed of at least two populations in the western U.S., and *C. posadasii* is composed of three populations widely dispersed across the American continents [7,8,9,10]. Given the high number of autapomorphic mutations between *Coccidioides* species and among isolates within species, variation in phenotypes is predicted [11]. However, minimal work characterizing phenotypic differences has been undertaken. A previous study demonstrated that *C. immitis* in vitro spherules grew in a synchronous pattern where *C. posadasii* isolates did not [12]. Differences in pathogenesis and other disease-associated phenotypic characteristics among strains have been reported, although only one study had species information [13,14,15,16,17,18]. The publication that defined the novel species *C. posadasii* also found species-specific variance in growth rate on media containing 0.136 M NaCl, suggesting that *C. immitis* is more salt tolerant than *C. posadasii*, but due to overlap in the phenotype, and evaluation of only 10 isolates of each species, it was not statistically meaningful [6]. These data supported observations published in the 1950–1960s, which proposed that salinity of the soil may be a factor in determining the distribution of *C. immitis* in Californian soil [19,20,21]. In contrast, a correlation of *C. posadasii* with saline soils was not observed in Arizona, where other associations were observed [22,23,24,25,26]. Importantly, recent modeling analysis predicts the future expansion of *Coccidioides* species in response to changing climate dynamics [27]. Therefore, a robust investigation of abiotic tolerances that may either limit or enhance distribution of *Coccidioides* is needed [1,28,29]. Such vital information could provide clues regarding the ecological niche, geographical range limits, or host-specific adaptations of the two species of *Coccidioides*.

The division of *Coccidioides* into two species has been challenged by clinicians because of the lack of apparent difference in disease manifestation caused by the two pathogens, but recent work suggests that there might be differences in dissemination patterns between the species [1,2,30]. Unfortunately, diagnosis and treatment of coccidioidomycosis does not require clinicians to identify to species. The current diagnostic methods, AccuProbe^®^ [31], CocciDx [32], and CocciENV [33], do not distinguish between the two species. Molecular-based technologies exist to differentiate the two species, but these have not been adapted to clinical use [34,35]. However, genotyping the causative agent would allow correlation of clinical presentations and outcomes associated with species. Severe disease and death typically occurs in high risk group patients; however, seemingly healthy individuals can succumb as well, without a known host immunologic or pathogen genotypic explanation [36]. Currently, the range of disease manifestations is suggested to be primarily due to host factors [37,38]. There are data supporting variation of virulence among individual isolates, but there is limited research on the subject [1,13,16,17,39]. A reasonable hypothesis would acknowledge that both host and pathogen genetics play a role in disease outcome and should be further investigated [40,41,42,43]. *Coccidioides*, like other human fungal pathogens, can grow at 37 °C—mammalian body temperature—which contributes to establishing host infection [44,45].

Thermotolerance is an intrinsic characteristic of an organism that allows for tolerance of excessively high temperatures. Heat acclimation can shape natural populations for a wide range of microorganisms, and is a physiological adaptation to heat stress imposed by the colonization of new habitats, global climate change and encountering new hosts [46,47,48,49,50,51,52,53,54]. This “preadaptation” is particularly important to pathogenic fungi that can grow in high temperatures, which allows colonization of mammalian tissues [55,56]. For example, *Coccidioides* is adapted to grow at high temperatures in the environment (i.e., North and South American deserts), and is able to colonize a wide range of endothermic hosts throughout the Americas [57,58,59,60,61]. *C. immitis* is endemic to the California Central Valley; whereas *C. posadasii* is widely distributed, but has highest prevalence in the Sonoran Desert. The annual mean temperature varies between the endemic areas, with the California Central Valley having more mild temperatures compared to the Sonoran Desert, which led us to hypothesize that *C. posadasii* is more thermotolerant than *C. immitis.* Therefore, we investigated the growth rate of both species at 37 °C and 28 °C, so that we might elucidate species-specific phenotypic variation. Here we demonstrate thermotolerance dissimilarity of the two species by analyzing growth rates of 85 isolates at these two temperatures.

## 2. Materials and Methods

Strains and Media. 39 *C. posadasii* strains and 46 *C. immitis* strains used in this study are primarily human patient isolates archived by various institutions, as detailed in Table 1 [6,8,28,62]. These strains represent both the full geographic range of the two species, and the proposed geographically distinct sub-populations [6,8]. Strains were grown on 2xGYE media (2% glucose, 1% yeast extract, 1.5% agar *w*/*v*) to supply initial plugs to inoculate plates for growth analysis, as this is the most common media used in *Coccidioides* laboratories. Yeast Extract (YE) media (0.5% yeast extract, 1.5% agar *w*/*v*) was used for growth experiments. Flagstaff Medical Center isolates were collected under IRB No. 764,034 through Northern Arizona Healthcare as part of the Northern Arizona University Biobank.

Growth Conditions and Measurements. Colonies were started by spreading approximately 10^6^ arthroconidia over the entire surface of a 2xGYE plate to create a lawn of mycelium to be transferred to initiate the thermotolerance experiment; this allowed measurement of colonial growth and not spore germination differences. After five days of growth at 25 °C, 7 mm diameter mycelial plugs were subcultured to the center of YE plates using a transfer tool (Transfertube^®^ Disposable Harvesters, Spectrum^®^ Laboratories, a Repligen Brand, CA, USA). Three replicates of each strain were plated for each experiment. All plates (100 mm × 15 mm BD Falcon 1015, Corning Life Sciences, MA, USA) were sealed with gas permeable seals (TimeMed Time^®^Tape, PDC Healthcare, CA, USA or Shrink Seals, Scientific Device Laboratory, IL, USA) for safety. Plates were placed in temperature-controlled incubators at either 28 °C or 37 °C in the dark under ambient humidity (30–50% RH) and CO_2_ (0.1%) conditions. Plate stacks were rotated from top to bottom and repositioned in the incubator with each measurement timepoint to reduce effects of environmental variation within the incubators. For measurement of radial growth, the diameter of each colony was measured in mm at 5, 7, 9, 12, 14, and 16 days post-subculture. The initial experiment occurred at University of Arizona (UA) in 2004, and subsequent testing with a new set of isolates occurred at Northern Arizona University (NAU) in 2019. Details for strains tested at each institution are listed in Table 1 and all raw measurement data are available in Appendix A.

Statistical Analysis. To estimate the mean growth rate for each species over the two-week period a mixed effect linear model for each temperature was constructed using the lme4 package in R version 3.6.2 [63,64]. Initially, data sets were divided by institution to conduct an analysis of variance; however, after concluding that species-specific growth rate was not strongly impacted by collection site the data sets were combined (Appendix A). In the temperature specific models, the factors “day” and “species” were assumed to be fixed linear effects, and individual isolate response for each day was considered to be a normally distributed random effect as appropriate in a longitudinal study. Thus, the response variable of colony diameter was modeled with fixed effects and a random effect to determine if growth rates varied between strains at either 28 °C or 37 °C. Shapiro-Wilk test (*p*-value < 0.001) shows that residuals are not normally distributed. However, the large sample size and overall residual structure support that a linear model is the most appropriate for this data set. In addition, bootstrapping using the boot package in R [65,66] was used to estimate 95% confidence intervals (CIs) for growth rates and other fixed effects (nsim = 2000). All bootstrap parameters were similar and support model estimates. A comparison between bootstrapped CIs and CIs constructed using the linear model can be found in Appendix A.

## 3. Results

To define variability of one phenotypic trait between two *Coccidioides* species, we examined the ability of *Coccidioides* spp. to grow in filamentous form at 37 °C and 28 °C on YE agar. In this study, we surveyed 85 strains of *Coccidioides*, representing isolates from the entire geographical range (Figure 1). Initial investigations occurred at the University of Arizona, and subsequent studies occurred at Northern Arizona University (Table 1).

We observed that mean growth rates varied between institutions; however, overall species-specific temperature behavior was comparable. Therefore, data sets were combined (Appendix A). Using a mixed effect linear model, we showed a significant species-specific difference for growth of the mycelial phase of the fungus based on temperature (Figure 2 and Table 2). Table 2 summarizes the estimated growth rate for each species, 95% confidence interval (CI), and *p*-value for each temperature specific model. Both species grew faster at 28 °C than 37 °C. Although *C. immitis* isolates had a larger mean diameter than *C. posadasii* on all days tested at 28 °C (Appendix A), the overall rate of increase was not statistically significant (*p*-value = 0.072, Table 2). This was in contrast to growth at 37 °C. At this temperature, *C. posadasii* strains exhibited larger mean diameters, which reached double the diameter of *C. immitis* by day 16 (Appendix A). At this temperature, the overall growth rate of *C. posadasii* was 1mm/day faster than *C. immitis* (Figure 2 and Table 2). This difference was statistically significant (*p*-value < 0.001, Table 2). These findings were consistent for all days tested and represent differential phenotypes for both species. Thus, our analysis indicates that high temperature is the important variable between species growth rate on solid media. This phenotypic difference supports the molecular phylogenetic species designation and may reflect adaptation of *C. immitis* to cooler environments, or possibly specific hosts.

Summary of temperature specific linear models, for 28 °C and 37 °C, respectively. Colony growth estimates for each species per day (slope), 95% confidence intervals (CI) and *p* values. At 28 °C, *C. immitis* grows 3.73 mm/day which is 0.26 mm faster per day than *C. posadasii*. The difference in slope is not significant (*p* = 0.072) based on α = 0.05. However, the *p*-value trends towards significance. At 37 °C, *C. immitis* grows 0.64mm/day which is 1.18mm slower than *C. posadasii.* The difference in slope (CI, 0.98–1.38 mm/day) is statistically significant (*p* < 0.001).

## 4. Discussion

Although many studies have looked at genetic variation among isolates of both species of *Coccidioides*, few studies have compared phenotypic differences. Observed genetic diversity between and within species makes it reasonable to hypothesize that phenotypic variation exists. We propose that a methodical documentation of phenotypic variation is a necessary first step to determine the ecological or clinical relevance of these traits. In this study, we have identified a definitive phenotypic difference with a congruent analysis at two institutions for a diverse set of isolates. A total of 85 isolates covering the geographic range of both species show that *C. posadasii* isolates grow at a significantly faster rate (*p* < 0.001, Figure 2 and Table 2) than *C. immitis* isolates in the mycelial form at 37 °C on YE agar. Additionally, *C. immitis* grows slightly faster than *C. posadasii* at 28 °C on YE agar although the difference in growth rate is not significant (*p*-value = 0.072, Figure 2 and Table 2). We note that growth rate may be influenced by nutrition source, and the results are limited to the media utilized for the current study.

Functionally, this phenotype is similar to a classic temperature sensitive (ts) conditional mutant, such that *C. immitis* exhibits normal growth at permissive temperature, and significantly slower growth under stressful conditions. It is possible that *C. immitis* could be restored to normal growth at 37 °C by gene replacement with appropriate *C. posadasii* alleles if candidate genes were identified. Currently, molecular techniques for genetic manipulations of *Coccidioides* are limited and need to be improved. Several genes and pathways have been described in *Aspergillus fumigatus* related to thermotolerance [54]. For example, the observed phenotype could be due to mutations in a heat shock protein (Hsp). Hsps are activated in response to changes in temperature and regulate cellular processes associated with morphogenesis, antifungal resistance, and virulence by triggering a wide array of cellular signaling pathways [53,67]. Hsps are activated by a heat shock transcription factor (Hsf) that acts as a thermosensor, regulating the Hsps at specific growth temperatures [68]. Several studies have shown that *Coccidioides* up-regulates heat shock proteins Hsp20 and Hsp9/12 at high temperature during the parasitic lifecycle while down-regulating Hsp30 and Hsp90 [69,70,71,72]. Further investigation of Hsps and Hsfs in *Coccidioides* could elucidate mechanisms of the species-specific thermotolerant behavior observed in this study. Alternatively, many classical ts mutants occur in genes required for normal cellular growth and are due to single amino acid changes that affect protein function or stability at the restrictive temperature. For example, a number of colonial temperature sensitive (*cot*) mutants have been identified in *Neurospora crassa*. The *N. crassa cot-1* mutant has been studied in greatest detail, and the ts defect is due to a SNP causing a single amino acid change in a Ser/Thr protein kinase required for normal hyphal extension, thus resulting in restricted growth at normally permissive temperatures above 32 °C [73,74]. Finally, recent work in *Saccharomyces* indicates that mitochondrial genotypes are associated with heat tolerance [75]. The mitochondrial genomes of the two species of *Coccidioides* are also distinct, and thus mitochondrial function is another potential mechanism controlling thermotolerance in *Coccidioides* [76].

The source of the genotypic variation driving the observed phenotype may be attributable to a stochastic event, such as a founder effect or population bottleneck 10–12 MYA, which is the estimated time that the two species have been separated [6,77]. Alternatively, the observed pattern may be due to selection pressure from a specific environment, host, or directly associated with virulence. In fact, the observed differential thermotolerance as tested in this investigation relates to the saprobic phase of the lifecycle and likely reflects adaptation to specific environments. A pattern of alternating wet–dry conditions has been related to Valley fever incidence across the southwestern U.S. [5,78,79,80,81,82]. It has been proposed that fungal growth occurs during brief periods of heavy moisture during monsoon and winter rainy seasons in the Southwest, which are followed by prolific conidia production when warm temperatures and low rainfall desiccate soils and increase dispersal via dust (i.e., the “grow and blow” hypothesis) [27,79,83]. Additionally, during high temperature periods, it is hypothesized that the surface soil is partially sterilized and many competitors are removed, but *Coccidioides* spores remain viable [26]. Another hypothesis is that *C. posadasii* may be better adapted to growth in the high soil temperatures observed in the southwestern deserts compared to the California endemic *C. immitis*. Maricopa, Pinal, and Pima counties harbor the highest coccidioidomycosis case rates in Arizona due to *C. posadasii*, and according to the National Centers for Environmental Information [84], the annual mean temperature (1901–2000) were 20.7 °C, 19.8 °C, and 19.2 °C, respectively. On the other hand, Fresno, King and Kern counties, which harbor the highest coccidioidomycosis case rates in California due to *C. immitis*, had annual mean temperatures of 12.4 °C, 16.9 °C, and 15.8 °C, respectively. The difference in 100-year average annual mean temperature between highly endemic areas of Arizona and California supports our hypothesis that *C. posadasii* is more adapted during saprobic growth to higher temperatures compared to *C. immitis*. Alternatively, a preferred host species may vary in normal body temperature, in accordance with the endozoan small mammal reservoir hypothesis proposed by Barker and Taylor [85]. Interestingly, a decline in mean human body temperature (~1.6%) has recently been reported [86]. Whether this impacts coccidioidomycosis rates is unknown.

Published literature to date suggests that disease outcomes are related primarily to host-specific factors [37,38,87], and certainly, host genetic background can impact disease progression. We propose that pathogen-specific variation may also contribute to capricious disease outcomes in coccidioidomycosis patients. Currently, species-specific virulence is not well-documented in *Coccidioides* research, but has been suggested [1,13]. This is in part due to the use of a few characterized laboratory strains of *Coccidioides* for most hypothesis testing, primarily strains Silveira, C735 and RS [70,87,88,89,90,91]. Therefore, connecting phenotypic dissimilarity to established genetic variation using genome-wide association studies could provide novel insight into unique characteristics of these genetically distinct pathogens.

## 5. Conclusions

In summary, we have identified a significant phenotypic difference between *C. immitis* and *C. posadasii.* Although growth rate on YE media at two temperatures is the only characteristic we explicitly tested, there are certain to be more phenotypic differences between species, and possibly between populations. This, coupled with the recent availability of the genome sequence of multiple strains for both fungal species, may allow comparative genomic approaches to elucidate candidate genes for thermotolerance regulation in *Coccidioides* and closely related Onygenales [7].

## Figures and Tables

**Figure 1 jof-06-00366-f001:**
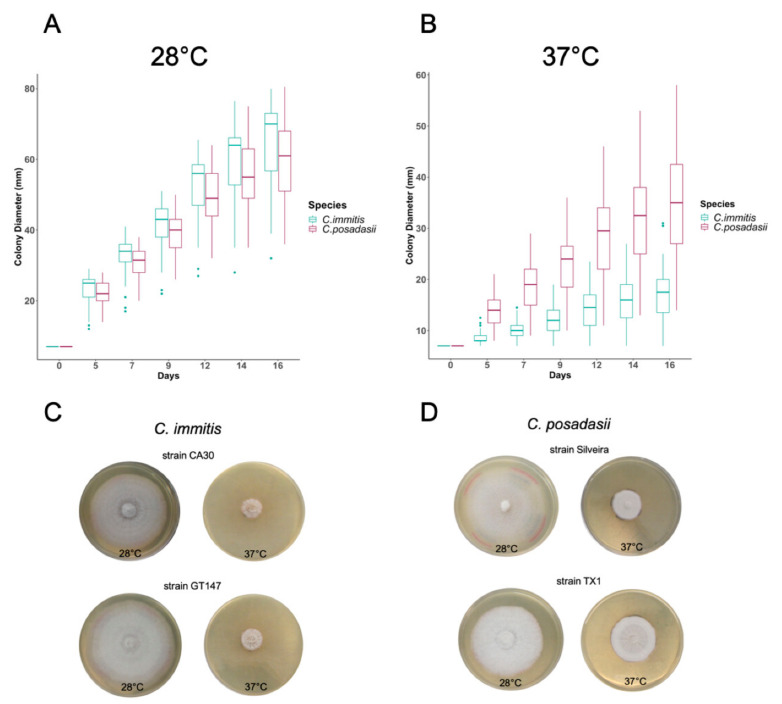
Temperature impacts growth ability of *C. immitis* isolates compared to *C. posadasii* on YE media. Seven mm diameter plugs were sub-cultured onto yeast extract plates and radial growth was documented over 16 days. (**A**) Radial growth measurements at 37 °C for 46 *C. posadasii* and 39 *C. immitis* isolates in triplicate. (**B**) Radial growth measurements at 28 °C for 46 *C. posadasii* and 39 *C. immitis* isolates in triplicate. (**C**) Representative of phenotypic variation observed for *C. immitis* on day 16. (**D**) Representative of phenotypic variation observed for *C. posadasii* on day 16.

**Figure 2 jof-06-00366-f002:**
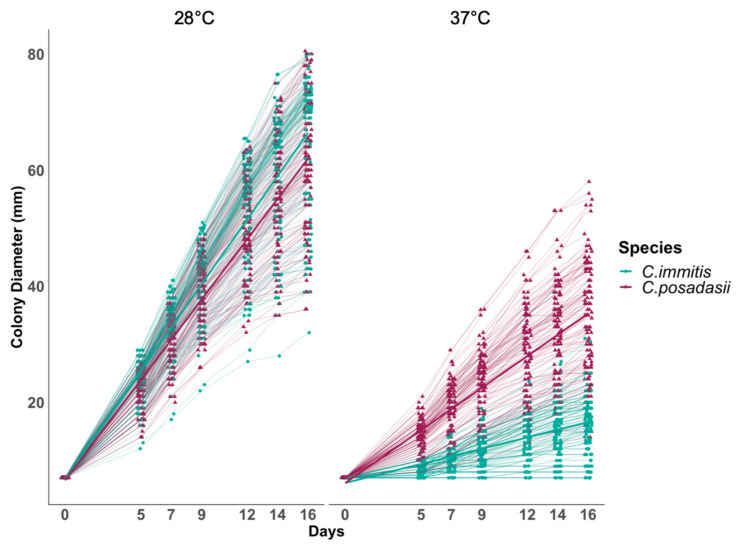
Radial growth rate of 85 isolates of *Coccidioides* demonstrates species-specific response to temperature. Each line represents the mean diameter (*y*-axis) for each isolate in triplicate (46 *C. immitis* and 39 *C. posadasii*) at a given time point (*x*-axis). Dark lines represent mean growth rate of each species. Radial growth was measured at day 5, 7, 9, 12, 14 and 16. There is a significant difference in growth rate (slope) in response to higher temperature between species of *Coccidioides*. The radial growth rate of *C. immitis* is decreased at a higher temperature 37 °C (slope37 = 0.64 mm/day; 95% C.I. 0.51–0.78) compared to *C. posadasii* (slope37 = 1.82 mm/day; 95% C.I. 1.49–2.16). Both species appear to tolerate 28 °C and grow at a similar rate (*C. immitis* slope28 = 3.73 mm/day; 95% C.I. 3.53–3.92, *C. posadasii*, slope28 = 3.47 mm/day; 95% C.I. 2.98–3.90).

**Table 1 jof-06-00366-t001:** Strain information.

ID	Species	NCBI Accession	Geographic Origin ^a^	Source	Testing Institution ^b^
CA22	*C. immitis*	NA	California	University of Texas Health Science Center (UTHSC)	NAU
500	*C. posadasii*	NA	Soil, Tucson, AZ	University of Arizona (UA)	UA
IL1	*C. posadasii*	NA	Illinois	UTHSC	NAU
CA23	*C. immitis*	NA	California	UTHSC	NAU
HS-I-000718	*C. posadasii*	NA	Arizona	Flagstaff Medical Center (FMC)	NAU
GT164	*C. posadasii*	NA	Texas	University of California Davis (UCD)	NAU
GT163	*C. immitis*	NA	California	UCD	NAU
HS-I-000588	*C. posadasii*	NA	Arizona	FMC	NAU
CA28	*C. immitis*	NA	California	UTHSC	NAU
TX4	*C. posadasii*	NA	Texas	UTHSC	NAU
HS-I-000235	*C. posadasii*	NA	Arizona	FMC	NAU
TX1	*C. posadasii*	NA	Texas	UTHSC	NAU
HS-I-000778	*C. posadasii*	NA	Arizona	FMC	NAU
GT147	*C. immitis*	NA	California	UCD	NAU
HS-I-000234	*C. posadasii*	NA	Texas	FMC	NAU
CA30	*C. immitis*	NA	California	UTHSC	NAU
HS-I-000547	*C. posadasii*	NA	Arizona	FMC	NAU
HS-I-000233	*C. posadasii*	NA	Arizona	FMC	NAU
GT166	*C. posadasii*	NA	Texas	UCD	NAU
CA24	*C. immitis*	NA	California	UTHSC	NAU
CA29	*C. immitis*	NA	California	UTHSC	NAU
M211	*C. posadasii*	NA	Central Mexico	Unidad de Micologia, UNAM	NAU
GT158	*C. posadasii*	NA	Arizona	UCD	NAU
CA15	*C. immitis*	NA	California	UTHSC	NAU
CA27	*C. immitis*	NA	California	UTHSC	NAU
TX3	*C. posadasii*	NA	Texas	UTHSC	NAU
CA20	*C. immitis*	NA	California	UTHSC	NAU
RS	*C. immitis*	AAEC00000000.3	California	Common Laboratory Strain	NAU
Silveira	*C. posadasii*	ABAI00000000.2	California	Common Laboratory Strain	NAU
RMSCC2378	*C. posadasii*	NA	Argentina	R. Negroni	UA
RMSCC2377	*C. posadasii*	NA	Argentina	R. Negroni	UA
RMSCC2379	*C. posadasii*	NA	Argentina	R. Negroni	UA
RMSCC3698	*C. immitis*	NA	Barstow, California	Naval Hospital	UA
RMSCC3490 ^c^	*C. posadasii*	SRR3468073	Coahuila, Mexico	I. Gutierrez	UA
RMSCC3505	*C. immitis*	NA	Coahuila, Mexico	I. Gutierrez	UA
RMSCC3506 ^c^	*C. posadasii*	SRR3468053	Coahuila, Mexico	I. Gutierrez	UA
RMSCC3472	*C. posadasii*	NA	Michoacán, Mexico	I. Gutierrez	UA
RMSCC3474	*C. immitis*	NA	Michoacán, Mexico	I. Gutierrez	UA
RMSCC3475	*C. immitis*	NA	Michoacán, Mexico	I. Gutierrez	UA
RMSCC3476 ^c^	*C. immitis*	SRR3468020	Michoacán, Mexico	I. Gutierrez	UA
RMSCC3478	*C. posadasii*	NA	Michoacán, Mexico	I. Gutierrez	UA
RMSCC3479 ^c^	*C. immitis*	SRR3468018	Michoacán, Mexico	I. Gutierrez	UA
RMSCC3377	*C. immitis*	NA	Monterey, California	UCD	UA
RMSCC2343 ^c^	*C. posadasii*	SRR3468064	Nuevo Leon, Mexico	R. Diaz	UA
RMSCC2346 ^c^	*C. posadasii*	SRR3468065	Nuevo Leon, Mexico	R. Diaz	UA
RMSCC3738	*C. posadasii*	NA	Piaui, Brazil	B. Wanke	UA
RMSCC3740	*C. posadasii*	NA	Piaui, Brazil	B. Wanke	UA
RMSCC2127	*C. posadasii*	NA	Texas	UTHSC	UA
RMSCC2133	*C. posadasii*	GCA_000150185.1	Texas	UTHSC	UA
RMSCC2234	*C. posadasii*	NA	Texas	UTHSC	UA
RMSCC2102	*C. immitis*	NA	San Diego, California	University of California San Diego (UCSD) Medical Center	UA
RMSCC2394	*C. immitis*	GCA_000149895.1	San Diego, California	UCSD Medical Center	UA
RMSCC2395	*C. immitis*	NA	San Diego, California	UCSD Medical Center	UA
RMSCC3693	*C. immitis*	NA	San Diego, California	Naval Hospital	UA
RMSCC3703	*C. immitis*	GCA_000150085.1	San Diego, California	UCSD Medical Center	UA
RMSCC3705	*C. immitis*	NA	San Diego, California	UCSD Medical Center	UA
RMSCC3706 ^c^	*C. immitis*	SRR3468019	San Diego, California	UCSD Medical Center	UA
RMSCC2006	*C. immitis*	NA	San Joaquin Valley	Kern County Public Health (KCPH)	UA
RMSCC2009 ^c^	*C. immitis*	SRR3468015	San Joaquin Valley	KCPH	UA
RMSCC2010	*C. immitis*	NA	San Joaquin Valley	KCPH	UA and NAU
RMSCC2011	*C. immitis*	NA	San Joaquin Valley	KCPH	UA
RMSCC2012 ^c^	*C. immitis*	SRR3468016	San Joaquin Valley	KCPH	UA
RMSCC2014	*C. immitis*	NA	San Joaquin Valley	KCPH	UA
RMSCC2015 ^c^	*C. immitis*	SRR3468027	San Joaquin Valley	KCPH	UA
RMSCC2017 ^c^	*C. immitis*	SRR3468038	San Joaquin Valley	KCPH	UA
RMSCC2268 ^c^	*C. immitis*	SRR3468049	San Joaquin Valley	KCPH	UA
RMSCC2269 ^c^	*C. immitis*	SRR3468060	San Joaquin Valley	KCPH	UA
RMSCC2271	*C. immitis*	NA	San Joaquin Valley	KCPH	UA
RMSCC2273 ^c^	*C. immitis*	SRR3468071	San Joaquin Valley	KCPH	UA
RMSCC2274	*C. immitis*	NA	San Joaquin Valley	KCPH	UA
RMSCC2275	*C. immitis*	NA	San Joaquin Valley	KCPH	UA
RMSCC2276	*C. immitis*	NA	San Joaquin Valley	KCPH	UA
RMSCC2277 ^c^	*C. immitis*	SRR3468079	San Joaquin Valley	KCPH	UA
RMSCC2278	*C. immitis*	NA	San Joaquin Valley	KCPH	UA
RMSCC2279 ^c^	*C. immitis*	SRR3468080	San Joaquin Valley	KCPH	UA
RMSCC2280 ^c^	*C. immitis*	SRR3468081	San Joaquin Valley	KCPH	UA
RMSCC2281 ^c^	*C. immitis*	SRR3468017	San Joaquin Valley	KCPH	UA
RMSCC3480 ^c^	*C. posadasii*	SRR3468051	Sonora, Mexico	I. Gutierrez	UA
RMSCC3487 ^c^	*C. posadasii*	SRR3468052	Sonora, Mexico	I. Gutierrez	UA
RMSCC3488	*C. posadasii*	GCA_000150055.1	Sonora, Mexico	I. Gutierrez	UA
RMSCC1040	*C. posadasii*	NA	Tucson, Arizona	UA	UA
RMSCC1043	*C. posadasii*	NA	Tucson, Arizona	UA	UA
RMSCC1044	*C. posadasii*	NA	Tucson, Arizona	UA	UA
RMSCC1045	*C. posadasii*	NA	Tucson, Arizona	UA	UA
RMSCC3796	*C. posadasii*	NA	Venezuela	G. San-Blas	

^a^ Often patient diagnosis location. ^b^ Northern Arizona University (NAU) University of Arizona (UA) ^c^ Isolate IDs were changed in Bioproject PRJNA274372 [7].

**Table 2 jof-06-00366-t002:** Temperature Specific Linear Model Slope Estimates for Radial Growth Rate at 28 °C or 37 °C.

	Colony Diameter at 28 °C			Colony Diameter at 37 °C		
Species	mm/Day	95% CI	P ^a^	mm/day	95% CI	P ^a^
*C. immitis* × Day	3.73	3.53–3.92	0.072	0.64	0.51–0.78	<0.001
*C. posadasii* × Day	3.47	0.55–0.02		1.82	0.98–1.38	
N ^b^	85			85		

^a^ difference between estimated slope, ^b^ number of isolates.

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
