# Peer review of "Differential Thermotolerance Adaptation between Species of Coccidioides"

_jof, 2020, doi:10.3390/jof6040366_

Round 1

Reviewer 1 Report

The paper by Mead et al. looks to expand on the knowledge in the coccidioidomycosis field; as the authors mention in the paper, molecular and genetic work has shown there are two distinct species in Coccidioides, but there is a lack of sound studies looking at phenotypic differences among the two species. This study by Mead et al. has conducted robust statistics analysis with a good number of isolates for the significant geographical areas where Coccidioides is found. Their finding clearly shows significant differences in the mycelial growth at 37°C between C. immitis and C. posadasii.

The paper's major issue is the lack of justification for only looking at the mycelial growth at 37°C. The authors need to address the clinical relevance of mycelia at 37°C. The concern is that once in the lungs, Coccidioides arthroconidia start the parasitic growth (Spherules).  Previously the group has quantified cell density and size of spherules (PMID: 30053114), suggesting that a comparison of the spherule growth at 37°C between the isolates used in this study is possible and should be included in order to advance the field forward. 

Minor comments:

  • Change the ordinal indicator symbol (º) for the degree symbol (°) in the abstract, materials and methods, results, Figure 1 and 2, and table 2.
  • In table 1, please specify at the bottom of the table NAU (Northern Arizona University), UA (University of Arizona)

Author Response

1.We respect the reviewers concern that filamentous growth at 37ºC may/may not have clinical relevance. We have changed language on line 105 and removed the discussion of host temperature to ensure focus on thermotolerant adaptions to the soil environment. Additionally, we mention on line 202-204 that a methodical documentation of variation is necessary to determine ecological or clinical relevance. This is the motivation of our paper.

Due to the limited documentation of phenotypic variation in Coccidioidies we believe it is important to publish our finding specific to mycelial growth rather than delaying to expand the project. Many clinical laboratories use 37ºC to culture pathogenic fungi (Aspergillus, Cryptococcus) as well as other dimorphic fungi (Histoplasma, Blastomyces, Paracoccidioides). These dimorphs convert to the parasitic form at 37ºC on solid media. While Coccidioides has a unique spherule structure at 37ºC this does not discount the phenotypic observations which reflect a species-specific behavior and evolutionary adaptation described in this manuscript. Screening 86 isolates for spherule growth is at a minimum a 2-year project. In order to start spherule cultures, 10^6 cell/ml of arthroconidia per isolate is required. Obtaining this high volume of arthroconidia takes 10-25 standard agar plates (depending on the strain) which need 4-6 weeks to develop and another 2 weeks to harvest and quantify. Addtioanly variation in both conidia production and spherule formation has been observed (see Friedman's papers from 1950s). Spherule morphogenesis is not completely understood, but is likely triggered by external cues beyond just temperature such as; oxygen, carbon dioxide, and nutrients. These factors must be fully explored, which would require multiple rounds of growth in different conditions. We agree this work should be completed and are actively working on a robust analysis of spherule development across the Coccidioides genus.

Minor comments:

  • Change the ordinal indicator symbol (º) for the degree symbol (°) in the abstract, materials and methods, results, Figure 1 and 2, and table 2.

All degree symbols have been corrected, thank you for pointing out this format error.

  • In table 1, please specify at the bottom of the table NAU (Northern Arizona University), UA (University of Arizona)

We have added the suggested footnote.

Reviewer 2 Report

This article titled, “Differential Thermotolerance Adaptation between species of Coccidioides” by Heather Mead et al. describes differential growth rate between C. posadasii and C. immitis. It is well written, the methods are sound, and the results are based on appropriate analyses. I have a few suggestions for improvement of the manuscript below. Of important note, this group is among the first to document phenotypic diversity at the inter species level in a phenotype of clinical relevance. The have referenced similar studies, mostly from the 50s, which document variation at the intra-specific and immunologic levels, but this the most recent, robust, and well powered documentation of phenotypic variation to date. This is significant because it will allow comparative genomics based approaches to unraveling the genetics of Coccidioides virulence and pathogenicity.

Comments/Questions:

The logic that fungi experience abiotic stresses in their native habitats which may shape virulence is a concept ahead of its time. Im very excited to see temperature dissected here and hopefully desiccation, smoke, UV, salt tolerance, and interactions with ameobae in following studies.

I have observed extreme variation in isolates’ morphologic based on where they are in the incubator. Therefor, I appreciate that the authors took care to randomize the plate placement between measurements.

How were the differential sample sizes taken into account?

Are these isolates genotyped? If so can we see accession numbers for genomes or genes?

Suggestions for improvement:

While GYE media is the standard in the field for growing Cocci hyphae, the manuscript would benefit from added commentary on why this media was selected and how it may or may not represent the soil environment.

The authors note that measurements of colony diameter were obtained at two locations, please specify how long between experiments these were collected. The due diligence was done to compare and evaluate statistically significant variation between means, and it is appropriate given the barriers to BSL3 research. However, the reader and field would benefit from knowing this information for future study design and data interpretation.

In the discussion section, it is states that documentation of variance is the first step in using genomics to dissect functionality of genes underlying phenotypes. What is coming next? I would love to see transcriptomic data, or other evaluations of genetic drivers for the phenotypic differences in the discussion.

Have the authors considered an allele swap experiment? It is hinted in the paragraph regarding heat shock proteins, but I wonder if the appropriate technology exists to do such an experiment.

Please add a reference to the statement on mitochondrial genetic variation on ling 233.

Author Response

How were the differential sample sizes taken into account?

The lme4 package uses a mixed effects linear model which takes into account unbalanced design. The citation for this package is included. We are not making inference or estimating confidence intervals about effects due to lab. Therefore, sample sizes does not impact our results. Because lab is included as a fixed effect the lme4 package interprets lab as a categorical variable independently from one another. So, while there are more samples included for UA the does not impact how they are compared to one another.

https://onlinelibrary.wiley.com/doi/full/10.1002/9781118445112.stat05514.

Are these isolates genotyped? If so can we see accession numbers for genomes or genes?

Including accession numbers is a great suggestion. We have added this information to table 1 where available.

Suggestions for improvement:

While GYE media is the standard in the field for growing Cocci hyphae, the manuscript would benefit from added commentary on why this media was selected and how it may or may not represent the soil environment.

We appreciate this recommendation. In lines of 237-241 of the discussion we addressed this as a limitation of this study.

The authors note that measurements of colony diameter were obtained at two locations, please specify how long between experiments these were collected. The due diligence was done to compare and evaluate statistically significant variation between means, and it is appropriate given the barriers to BSL3 research. However, the reader and field would benefit from knowing this information for future study design and data interpretation.

We have included this information in the methods section on line 131-132.

In the discussion section, it is states that documentation of variance is the first step in using genomics to dissect functionality of genes underlying phenotypes. What is coming next? I would love to see transcriptomic data, or other evaluations of genetic drivers for the phenotypic differences in the discussion.

As hinted in the discussion we are very interested in the genetic characteristics responsible for the described phenotype. We and others in the Coccidioides community are actively working on high quality genome assemblies, gene annotation and genotyping of disparate strains within each genus. These foundational steps will allow us to ask sophisticated questions in the future.

Have the authors considered an allele swap experiment? It is hinted in the paragraph regarding heat shock proteins, but I wonder if the appropriate technology exists to do such an experiment.

Yes, we have considered an allele swap experiment, or creating mutant libraries in a C. posadasii background then screening for C. immitis-like thermotolerance phenotypes and the genetic associated mutations. The precise technology required for this work does not yet exist as gene editing strategies in Coccidioidies are poorly developed and often result in heterokaryons or include off-target effects  We are actively developing CRISPR Cas9 technology for precise gene editing in Coccidioides to facilitate future investigations.

Please add a reference to the statement on mitochondrial genetic variation on ling 233.

The citation for the mitochondrial reference has been included.